# Emerging Roles of Small Extracellular Vesicles in Gastrointestinal Cancer Research and Therapy

**DOI:** 10.3390/cancers16030567

**Published:** 2024-01-29

**Authors:** Nora Schneider, Patrick Christian Hermann, Tim Eiseler, Thomas Seufferlein

**Affiliations:** Department for Internal Medicine 1, University Clinic Ulm, 89081 Ulm, Germany; patrick.hermann@uni-ulm.de (P.C.H.); thomas.seufferlein@uniklinik-ulm.de (T.S.)

**Keywords:** gastrointestinal cancer, small extracellular vesicles, sEVs, metastasis, pre-metastatic niches, biomarkers, therapeutic sEVs

## Abstract

**Simple Summary:**

Gastrointestinal cancers originate in the digestive system and harbor distinct characteristics according to their site of origin. This review focusses on the role of small extracellular vesicles (sEVs, exosomes) in the progression, metastasis, and treatment of the main GI cancer entities, such as colorectal cancer, gastric cancer, hepatocellular carcinoma, and pancreatic ductal adenocarcinoma. In recent years, sEVs have gained increasing attention as important mediators of intercellular communication within the local tumor microenvironment, and also to distant metastatic niches. sEVs deliver bioactive cargos, like proteins, mRNA, and miRNAs, to reprogram target cells, promoting tumor growth, invasion, immune suppression, and metastasis to specific organs. Due to their presence in all biological fluids, sEVs are ideal biomarker platforms for multiplexing analysis. Furthermore, sEV engineering generated promising approaches for the use of sEV-based therapeutic nanovesicles in GI cancer treatment.

**Abstract:**

Discovered in the late eighties, sEVs are small extracellular nanovesicles (30–150 nm diameter) that gained increasing attention due to their profound roles in cancer, immunology, and therapeutic approaches. They were initially described as cellular waste bins; however, in recent years, sEVs have become known as important mediators of intercellular communication. They are secreted from cells in substantial amounts and exert their influence on recipient cells by signaling through cell surface receptors or transferring cargos, such as proteins, RNAs, miRNAs, or lipids. A key role of sEVs in cancer is immune modulation, as well as pro-invasive signaling and formation of pre-metastatic niches. sEVs are ideal biomarker platforms, and can be engineered as drug carriers or anti-cancer vaccines. Thus, sEVs further provide novel avenues for cancer diagnosis and treatment. This review will focus on the role of sEVs in GI-oncology and delineate their functions in cancer progression, diagnosis, and therapeutic use.

## 1. sEVs–Biogenesis, Cargo Loading, Secretion, and Uptake

Extracellular vesicles (EVs) are lipid bilayer nanovesicles that are physiologically released from almost all cell types. Based on their respective size and distinct biogenesis pathways, major subclasses can be defined, such as apoptotic bodies, which are the largest EVs with a size greater than 1000 nm, microvesicles (100–1000 nm), and small extracellular vesicles (sEVs, exosomes) with a diameter of 30–150 nm [1,2,3]. The International Society for Extracellular Vesicles (ISEV) has also published a statement on the minimal information that defines the following parameters for sEVs required for experimental studies: Size distribution 30–150 nm, as defined by nano-particle tracking analysis (NTA) or dynamic light scattering (DLS), detection of sEV surface markers, like tetraspanins (CD63, CD81, CD9), measurement of luminal markers (e.g., TSG101), exclusion of endosomal vesicle contaminations by GRP94, as well as demonstration of nanovesicles by transmission electron microscopy (TEM) [4]. This review will focus on small extracellular vesicles (sEVs) in GI cancer entities, and will use the term “sEVs” instead of exosomes to allow for easy communication with the reader [5].

sEVs were initially discovered in 1981 by Trams et al. as exfoliated membrane vesicles of red blood cells [6], and they were thought to function as cellular waste bins. However, in recent years, the focus has shifted to intercellular communication as a key function of sEVs. This is mediated by transmission or signaling through bioactive cargos, which include lipids, proteins, metabolites, DNA, or different RNA classes such as full-length mRNA, miRNA, or lncRNA. This cargo reprograms recipient cells, both under healthy conditions and also in disease states, such as cancer [1,7]. In line, cancer cells release a large amount of sEVs, which fulfill important functions both in shaping the immediate tumor microenvironment (TME), and also in forming distant pre-metastatic niches to promote tumor growth and cancer dissemination [8,9,10].

sEVs also contribute to the regulation of the immune system and can be exploited by tumor cells to modulate and dampen an anti-tumor immune response that is mounted to a large extent by the adaptive immune system with CD8+ T-cells. The role of sEVs in modulating this process is still under investigation, but there is crucial evidence that sEVs regulate antigen processing in antigen-presenting cells (APCs), and also that they act directly on T-cells by altering their priming, activation, expansion, and survival [11]. Since sEVs can be detected ubiquitously in all body fluids, they have also been proposed as an ideal biomarker platform to detect various diagnostic or prognostic markers, such as microRNAs (miRNAs) or proteins, even utilizing multiplex approaches [9,12]. To this end, non-invasive liquid biopsy methods—e.g., analyzing sEVs in patient blood samples—are increasingly used for biomarker studies [9].

### 1.1. Biogenesis Mechanisms

There are several pathways by which sEVs are formed and released from cells. The most well-known pathway involves the endosomal sorting complexes required for transport (ESCRT) machinery, which sorts tagged cargo into intraluminal vesicles (ILVs) within endosomal structures, so-called multivesicular bodies (MVBs) that fuse with the plasma membrane to release intraluminal nanovesicles as sEVs [1,8,9]. However, other pathways—such as ESCRT-independent mechanisms—have also been described. Each of the different biogenesis pathways has unique characteristics and is specifically regulated to generate distinct sEV sub-populations. The most extensively studied pathways are described in the following section in more detail (Figure 1).

The ESCRT machinery comprises a set of four protein complexes (ESCRT-0, ESCRT-I, ESCRT-II, and ESCRT-III) that work sequentially to sort ubiquitinated cargo into ILVs within MVBs. Subsequently, the MVBs containing ILVs are transported via microtubules to the plasma membrane where they release the sEVs. It can be shown that even after silencing key ESCRT subunits, the formation and release of MVB is still possible, indicating that ESCRT-independent ways of sEV biogenesis are existing as well [2,8,9,13].

In vitro experiments showed a decrease in sEV release when neutral sphingomyelinase (nSMase) is inhibited, indicating that ceramide is an important regulator of exosome biogenesis. The lipid-based biogenesis is driven by ceramides, lysophospho-, or glycosphingolipids, which incorporate into the membrane and thereby enable spontaneous budding of ILVs. The enzymatic conversion of ceramides to sphingosine and sphingosin1-phosphate (S1P) also activates sphingosine1-phosphate receptors on limiting membranes, which are implicated in the sorting of tetraspanins into ILVs [14]. Tetraspanins are major sEV biomarkers that regulate the clustering of surface proteins, such as integrins, endocytosis of the respective cargos, and distribution into intraluminal vesicles during biogenesis [1,8,9,13] (Figure 1). Another pathway involved in ESCRT-independent biogenesis is controlled by the small integral membrane protein of the lysosome/late endosome (SIMPLE), which was shown to increase sEV release upon transfection of COS cells [15].

### 1.2. Major Cargos and sEV Markers

Several proteins have been identified as specific markers of sEVs, which distinguish them from other extracellular vesicles and cell debris. Some major sEV markers are the tetraspanins CD9, CD63, CD81, CD82, CD53, and CD37. sEVs also contain other biogenesis related proteins, such as ALG-2 interacting protein X (ALIX) and tumor susceptibility gene 101 protein (TSG101), as well as Ras associated binding protein (Rab)–GTPases, controlling the transport and the release of sEVs from MVBs at the plasma membrane. Additionally, major surface markers such as class 1 and 2 major histocompatibility complexes (MHC I/II), annexins, flotillin (FLOT1), and integrins can be integrated into sEV membranes [8,9,13,16].

Other cargo classes include lipids and nucleic acids, such as DNA, mRNA, and miRNAs, as well as long noncoding or circular RNAs. The loading of RNA cargos into sEVs involves several mechanisms. One major factor is the recruitment of different RNA species by RNA binding proteins, like heterogeneous nuclear ribonucleoprotein A2/B1 (hnRNPA2B1) [8,17]. Additionally, some studies have suggested that RNA molecule loading is facilitated by specific sequence motifs in their respective RNAs (EXOmotif and EXO-SEQUENCE) [18]. A study by Jeppesen et al. has analyzed major protein, RNA, and DNA constitutes of sEVs. They further found that extracellular RNA and RNA binding proteins are differentially expressed in sEVs and non-vesicular compartments. Moreover, they state that Argonaute 1–4, as well as glycolytic enzymes and cytoskeletal proteins, are absent from sEVs [19]. The presence of DNA cargo molecules has been reported in sEVs [20] as well, although this is controversially discussed. To this end, Jeppesen et al. claim that sEVs are not vehicles of active DNA release, and that active secretion of extracellular DNA is thought to be an sEV-independent mechanism driven by an autophagy- and multivesicular endosome-dependent mechanism [19]. Proteins are sorted into sEVs via the ESCRT-machinery or by ESCRT-independent pathways—e.g., in a tetraspanin-dependent manner [8,16,21]. Additionally, some proteins are loaded into sEVs via interactions with cytosolic chaperones, such as heat shock proteins (HSP), which facilitate their incorporation into ILVs [22]. The lipid composition of sEVs is modulated by a combination of passive diffusion and active transport mechanisms. One example is the transfer of lipid raft domains from the plasma membrane into ILVs [23,24]. Another mechanism involves the interaction of specific lipid-binding proteins with the ESCRT machinery [8,9,25,26].

### 1.3. sEV Secretion

The final release of sEVs includes additional stages following MVB biogenesis. MVBs are transported along microtubules to the plasma membrane. To this end, Rab family GTPases, like Rab27a/b or Rab11, control various aspects of this transport process [27,28]. Subsequently, MVBs fuse with the plasma membrane, which is mediated by soluble N-ethylmaleimide-sensitive factor attachment protein receptors (SNAREs) [1,13,29]. In addition, the presence of branched actin filaments at the plasma membrane formed by the actin-related protein (Arp)-complex is required for efficient sEV release. Weaver and colleagues, as well as our group, have recently shown that this process is controlled by Cortactin [21,30,31], a downstream target of protein kinase D1 (PRKD1) [21]. Our data further indicate that the nucleation promoting factor WASp Family Verprolin-homologous Protein-2 (WAVE2) is required in this process to activate the Arp-complex [21].

Inducible secretion of sEVs is also dependent on several exogenous factors, like cellular stresses, including low pH, DNA damage, hypoxia, or increased intracellular Ca^2+^ concentration [13,32,33,34,35]. Moreover, major tumor driver mutations, such as Kirsten rat sarcoma virus (KRAS) or tumor suppressor protein 53 (Tp53) have been described to modulate both sEV secretion and cargo content [36,37,38,39]. In addition, altered intracellular signaling of the RAS-mitogen-activated protein kinase (MAPK) pathway, the Phosphoinositide 3-kinase-AKT pathway (PI3K-AKT), and the Mammalian target of rapamycin (mTOR) pathway, also translate into changes in sEV release [40,41]. These factors work in concert to regulate the quantity and quality of sEV secretion depending on the cell type, the physiological state, and the environmental context [8,13,25].

### 1.4. sEV Uptake in Target Cells

After release into the extracellular space or circulation, sEVs affect target cells through different mechanisms, including direct binding of cellular surface receptors or incorporation of sEVs and cargo transfer [1]. Uptake mechanisms utilize direct membrane fusion, clathrin-, or lipid raft (caveolae/caveolin-1)-mediated endocytosis, macropinocytosis, and phagocytosis [1,42,43,44,45,46]. Interestingly, different uptake mechanisms are prevalent in different tumor types. In the cancer context, uptake of sEVs supports different aspects of tumor growth, cancer progression, and metastasis [21]. sEV uptake can be detected using reporter systems, such as fluorescent labeled lipophilic dyes, labeled RNA, or proteins [21,47,48].

### 1.5. Inhibition of sEV Release and Uptake

The modulation of both sEV secretion and uptake are attractive approaches to interfere with intercellular communication in diseases such as cancer. However, the currently used sEV biogenesis and uptake inhibitors are not FDA/EMA-approved, or are not suitable for clinical use due to high dosage requirements or side effects. A common sEV secretion inhibitor for the ESCRT-independent pathway is the sphingomyelinase (nSMase) inhibitor GW4869, which inhibits ceramide release from sphingomyelin. Ceramide is required for ESCRT-independent sEV biogenesis by generating lipid raft domains involved in sEV shedding. ESCRT-dependent biogenesis of sEVs is inhibited by manumycin A, which blocks Ras farnesyltransferase activity and thus Ras activation. To this end, Datta et al. have reported that the Ras-dependent inhibition of sEV biogenesis is mediated by ERK-dependent inhibition of the oncogenic splicing factor hnRNP H1 [49]. Furthermore, nexinhib and macropinocyt target RAB27A, a protein involved in sEV release at the plasma membrane [49,50,51,52,53,54,55]. Uptake inhibition can be accomplished by high-dose heparin, which blocks the binding of heparin sulphate proteoglycans, and is thus thought to inhibit endocytosis [56]. Dynamin-2, a small GTPase required for pinching of membranes, is targeted by dynasore, inhibiting endocytosis [57]. Moreover, blocking macropinocytosis—e.g., by amiloride—also prevents sEV uptake [56,57,58,59,60,61]. Efforts to identify possible therapeutic uptake inhibitors, especially in cancer, are ongoing. Due to unspecific blocking of healthy sEVs and potential side effects, a therapeutic strategy to inhibit sEV uptake is still under development.

## 2. Functions of sEVs in Gastrointestinal Cancers

Gastrointestinal cancers (GI-cancers) originate in the digestive system, which includes the esophagus, stomach, small intestine, colon, rectum, liver, and pancreas. Those cancer entities have different characteristics; some are known for their very poor 5-year survival rate, and others respond quite sufficiently to chemotherapy, but all have a strong impact on a patient’s life expectancy and quality of life [62,63,64,65] In this review, we focus on the following main GI cancer entities: Gastric, colorectal, pancreatic, and liver cancer.

Gastric cancer (GC) is the fourth most common cause of cancer-related death worldwide, and is often diagnosed at an advanced stage. It is a multifactorial disease with common risk factors, such as Helicobacter pylori infection, smoking, alcohol consumption, and high salt intake [62,66]. Although the median age at diagnosis is around 70 years, the incidence, particularly among young adults (age < 50 years), has dramatically increased over the last years. For early gastric cancer, endoscopic resection is possible, whereas radical surgery for locally advanced tumors is recommended [62,66,67]. At advanced, metastatic stages, systemic chemotherapy is the primary treatment option. In addition, targeted therapeutic approaches (e.g., targeting human epidermal growth factor receptor 2 (HER2), Claudin 18-2 or tumor associated vessels), immunotherapy (e.g., anti-programmed death protein 1/ligand 1 (PD1/PD-L1), or anti-cytotoxic T-lymphocyte-associated protein 4 (CTLA4) monoclonal antibodies (mAbs) significantly improved the outcome of gastric cancer patients in recent years [62,68,69].

For colorectal cancer (CRC), the incidence in younger patients (early-on-set CRC) is dramatically increasing. Due to organized screening programs, early detection with endoscopic or surgical excision is possible, which has significantly reduced mortality over recent years in the screening population [70,71]. However, CRC is still frequently detected only at an advanced stage, and in some cases, a synchronous metastatic spread, preferentially to the liver, is present at the time of diagnosis [72]. A better pathophysiologic understanding and molecular characterization of the tumor led to groundbreaking advances in the systemic treatment of metastatic CRC. Aside from the RAS mutation status, the serine/threonine-protein kinase B-Raf (BRAF) and HER2 mutation status and the microsatellite instability status deliver both prognostic information as well as new treatment options [65].

The liver displays the sixth most common side of primary cancers, with hepatocellular carcinoma (HCC) accounting for 80–90% of all primary liver tumors. Chronic inflammation and cirrhosis constitute the most common background for the development of HCC [73]. If diagnosed at an early stage, surgery with curative intention is possible. Furthermore, other local treatment opportunities—e.g., ablative strategies or embolization—show promising therapeutic results. For systemic treatment of advanced stages, the combination of immunotherapy and antiangiogenic treatment—e.g., Atezolizumab and Bevacizumab or dual checkpoint blockade (using durvalumab plus termelimumab)—emerged as first-line treatments and showed superior efficacy compared to the old standard treatment with kinase inhibitors [64,74].

Of all GI-cancers, pancreatic ductal adenocarcinoma (PDAC) is known for its aggressiveness, therapy resistance, and poorest overall 5-year survival rate, at around 12%. The tumor is often detected at an advanced, metastatic state when only palliative chemotherapy with FOLFIRINOX (5-fluorouracil [5-FU], leucovorin, irinotecan, and oxaliplatin) or gemcitabine plus (nab)-paclitaxel can prolong survival [63]. Unfortunately, in most cases, the response to chemotherapy remains limited due to various resistance mechanisms of the tumor. A better molecular understanding of both the tumor microenvironment and the cellular subpopulations of PDAC could deliver new treatment options, and personalized medicine approaches are urgently needed to improve survival [63,75].

In the next chapter, we will summarize the current state of research in the respective fields of GI cancer, focusing on the role of sEVs as key mediators of intercellular communication during cancer progression and metastasis. We will also summarize the respective functions as biomarker platforms and therapeutic vehicles for novel treatment approaches (Figure 2).

### 2.1. Role of sEVs in Tumor Growth, Cancer Progression, and Chemoresistance

#### 2.1.1. Gastric Cancer

sEVs derived from gastric cancer promote tumor cell proliferation by activation of PI3K/Akt and MAPK [76]. In addition, MAPK-signaling can be induced by sEVs released from CD97-high expressing gastric cancer cells [77]. Gastric cancer proliferation and migration is controlled via transfer of the LncRNA ZFAS1 via sEVs [78]. Moreover, chemoresistance in gastric cancer can be conferred by sEVs from M2 polarized macrophages via transfer of miRNA-21 (miR-21) [79], promoting tumor cell survival. A major chemotherapeutic agent in gastric cancer is paclitaxel. Gastric cancer cell lines with paclitaxel-resistance were shown to transfer mir-155p via sEVs to sensitive cells and induce chemoresistance, likely by targeting GATA binding protein 3 (GATA) and tumor protein p53 inducible nuclear protein 1 (TP53INP1) [80].

#### 2.1.2. Colorectal Cancer

CRC cells were shown to drive proliferation by sorting the tumor-suppressive miRNA-193a into sEVs as part of a mechanism to eliminate unwanted tumor-suppressive compounds [81]. To this end, an antiproliferative effect of miR193a was demonstrated by targeting Caprin1, which is a positive regulator of cell cycle progression. In addition, CRC progression is controlled by sEV-based signaling [82]. sEVs from KRAS-mutated colon tumors enhance invasiveness of recipient cells in vitro by transferring tumor promotors, such as mutant KRAS [83], endodermal growth factor receptor (EGFR), and integrins to KRAS wildtype cells [84]. CRC-sEVs also induce epithelial-to-mesenchymal-transition (EMT), promoting invasiveness and loss of epithelial characteristics via transfer of miR-210 [85]. Despite improvements in targeted treatments and immunotherapy, chemotherapy still is a mainstay in CRC treatment. sEVs secreted from cancer-associated fibroblasts (CAFs) promote chemoresistance in CRC by enhancing both stemness and EMT [86]. Moreover, CAF-derived sEVs were described to deliver lncRNA19 to CRC cells, inducing stemness properties and drug-resistance by activating Wnt and beta-catenin [87].

#### 2.1.3. Hepatocellular Carcinoma

sEVs significantly affect tumor progression of HCC. To this end, hepatic stellate cells (HSC) in the tumor microenvironment (TME) secrete sEVs containing miR-21, which target PTEN and AKT-signaling in quiescent hepatic stellate cells [88]. These activated CAFs in turn promote tumor progression by secreting angiogenetic cytokines and EMT regulators, such as vascular endothelial growth factor (VEGF), matrix metalloproteinase 2 (MMP2), MMP9, basic fibroblast growth factor (bFGF), and transforming growth factor beta (TGF-beta) [89]. In addition, HCC-derived sEVs were reported to mediate HCC progression and recurrence by inducing EMT through MAPK/extracellular signal regulated kinase (ERK) signaling [90], while another study has shown EMT regulation through TGF-beta/SMAD signaling [91]. Although there are many treatment options in HCC, drug resistance is a challenging issue. In particular, sEVs from HCC were shown to induce sorafenib (TKI) resistance in vitro by activating hepatocyte growth factor/mesenchymal- epithelial transition factor/AKT (HGF/c-Met/Akt) signaling in other HCC cancer cells, thus preventing sorafenib-induced apoptosis [92].

#### 2.1.4. Pancreatic Cancer

The PDAC TME comprises immune cells, fibroblasts, myofibroblasts, stellate cells, and a large amount of dense extracellular matrix (ECM). CAFs remodel the ECM and support tumor growth [9]. In vitro studies have shown that sEVs released from gemcitabine exposed CAFs increase proliferation and survival of chemosensitive and chemoresistant PDAC cell lines via regulation of the transcription factors SNAIL and miR-146a [93]. Inhibition of sEV release in turn reduces PDAC cell proliferation and survival. In addition, sEVs from CAFs were shown to rescue proliferation of nutrient-deprived PDAC cells by supplying vital metabolites [9]. Tumor progression via EMT towards an invasive phenotype is also promoted by PDAC-derived sEVs. Here, tumor sEVs containing tenascin-C (TNC) were reported to induce Wnt/β-catenin signaling, EMT, and tumor progression [9]. sEVs also play a major role in chemoresistance [63]. Gemcitabine resistance was induced by exosomal miRNA-106b, released from CAFs upon treatment against TP53INP1 [94]. Furthermore, sEVs from Gemcitabin-resistant PDAC stem cells transfer drug resistance to gemcitabine-sensitive PDAC cells by delivering miR-210, which targets the mammalian target of rapamycin (mTOR) signaling [95].

### 2.2. Role of sEVs in Metastasis and Pre-Metastatic Niche Preparation (PMN)

#### 2.2.1. Gastric Cancer

Apart from lymph nodes, the liver is a major metastatic site in GC [96]. Zhang et al. demonstrated that EGFR in GC-derived sEVs is transferred to liver stromal cells, thereby suppressing miR-26a/b, which causes an upregulation of hepatocyte growth factor (HGF) expression. Paracrine secretion of HGF in turn mediates the interaction with migrated cancer cells via binding c-Met, thereby establishing the GC metastatic niche [97]. A second major manifestation for GC spread is the peritoneal cavity [96]. There, peritoneal mesothelial cells (PMCs) experience mesothelial-to-mesenchymal transition (MMT) to establish a favorable metastatic niche environment. This phenotype can be induced by transfer of GC-derived exosomal miR-21–5p by activating the TGF-β/Smad pathway [98].

#### 2.2.2. Colorectal Cancer

In CRC, lymph node metastasis is a prognostic factor in determining the overall 5-year survival of patients, since it is a predisposing factor for distant tumor dissemination [99]. Lymphatic vessels are generated by lymph angiogenesis, which can be initiated downstream of vascular endothelial growth factor (VEGF)-C and VEGF-D signaling [100]. Sun et al. demonstrated that the formation of lymphatic networks is promoted by CRC-derived sEVs, inducing VEGF-C signaling by macrophages in the sentinel lymph node via interferon regulatory factor 2 (IRF-2) containing sEV cargo [101]. In addition, CRC liver metastasis can be established by polarizing liver macrophages towards a pro-inflammatory, interleukin-6 (IL-6) secreting phenotype via transfer of miR-21 by CRC-derived sEVs [102]. This study underlines the role of CRC-derived sEVs in establishing premetastatic niche formation via pro-inflammatory signaling.

#### 2.2.3. Hepatocellular Carcinoma

HCC is the most common liver cancer. At an advanced stage, patients often present with lung metastasis [103]. Mao et al. found that pulmonary metastasis of HCC is enhanced by angiogenesis and pulmonary endothelial permeability driven by Nidogen-1-positive HCC-derived sEVs, thereby inducing pre-metastatic niche formation in the lung [104]. Additionally, miR210 was detected in serum sEVs of HCC patients, and it induced angiogenesis in vitro by targeting SMAD4, the signal transducer, and the activator of transcription 6 (STAT6) signaling, emphasizing the important role of angiogenesis in HCC dissemination [105].

#### 2.2.4. Pancreatic Cancer

PDAC is characterized by early metastasis, which is often already present at the time of the initial diagnosis. The liver and lungs, as well as the peritoneal cavity are the main sites for PDAC metastasis. Formation of distant metastatic sites depends on the establishment of pre-metastatic niches, supporting the survival and growth of cancer-initiating cells. In recent years, it has become evident that sEVs are vital communicators during the formation of pre-metastatic niches in specific organs (organotropic metastasis) [21,106]. To this end, specific integrin combinations on tumor-derived sEVs, such as αvβ5, α6β4-, or α6β1 were described to drive niche formation in the liver or lung, respectively [21,106,107]. Integrins are important signaling mediators during metastasis, drive cell-ECM adhesion, and cell motility. Interestingly, integrins from the cell surface are packaged into sEVs in a tetraspanin-dependent manner to regulate organotropic metastasis [21,106]. For the liver metastatic sites, Costa-Silva et al. found that PDAC-derived sEVs transfer the migration inhibitory factor (MIF) to Kupffer cells (KCs) in the liver, which in turn release TGF-β to facilitate pre-metastatic niche formation by hepatic stellate cells. In line with that, PDAC patients present with elevated levels of MIF-positive circulating sEVs compared to healthy control subjects. Distant metastasis of PDAC to the lungs is regulated by sEVs with the distinct integrin pattern, α6β4/1 [107]. For organotropic lung metastasis, we could show that α6β4 expression on PDAC-derived sEVs is regulated downstream of a signaling pathway initiated by the loss of a kinase, PRDK-1, via epigenetic mechanisms in aggressive PDAC [21]. Loss or inhibition of PRKD-1 strongly enhanced sEV release with altered integrin α6β4 surface cargo, determining metastasis to the lung, while integrin β5 was downregulated, leading to impaired liver metastasis. The in vitro data could be validated by injection of the respective sEVs in xenografted mice in vivo. Enhanced expression of integrin α6β4 on the respective sEVs was caused by transcriptional upregulation in cells as well as increased endosomal recycling and packaging in a tetraspanin CD82-dependent manner. Consequently, targeting CD82 impaired packaging of integrin α6β4 in their respective sEVs. The final establishment of pre-metastatic lung niches was mediated by lung fibroblasts and induction of S100A6, A13, and A16 expression [21]. Thus, targeting integrin cargo-packaging into sEVs may constitute an attractive new therapeutic approach for PDAC.

### 2.3. Functions of sEVs in Immune Modulation and Tumor Immune Escape

An important role of tumor-derived sEVs is immune modulation to target the anti-tumor immune response, both in the primary tumor matrix and at the distant pre-metastatic niches supporting tumor-dissemination [108]. Anti-tumor immunity is triggered by tumor-associated antigens, and results in the activation of innate and adaptive effector cells, such as natural killer cells and CD8+ T-cells, which directly eliminate tumor cells upon activation. They are also targeted by tumor-derived sEVs to facilitate immune escape. In GI cancer, sEVs promote immune evasion by reprogramming, suppressing, or killing immune cells; e.g., via expression pro-apoptotic Fas-ligands or immune-checkpoint regulators, like PDL-1 [109,110].

#### 2.3.1. Gastric Cancer

In GC, sEV-resident PDL-1 was associated with high-immunosuppressive activity and a poor prognosis [111]. Another study demonstrated that 5-FU chemotherapy induces upregulation of exosomal PDL-1 via miR-940, thus triggering the casitas B lymphoma-b (Cbl-b)/STAT5A axis to induce systemic immune suppression. The respective sEVs induced apoptosis in Jurkat T-cells and prevented T-cell activation in peripheral blood mononuclear cells (PBMCs) [112]. This study indicates that chemotherapy-induced signaling can severely impact anti-tumor immunity via intercellular communication by tumor-derived sEVs, not only at local tumor sites, but also with systemic consequences.

#### 2.3.2. Colorectal Cancer

Huber et al. have shown that CRC cell-derived EVs contain Fas-Ligand and tumor necrosis factor-related apoptosis-inducing ligand (TRAIL), thereby inducing apoptosis of T-cells in vitro and in vivo [113,114]. One of the most abundant cell types in the CRC environment are tumor-associated macrophages (TAMs). A recent study demonstrated that specific CRC-cancer derived sEV-miRNAs induce macrophage M2-polarization and PDL-1 expression via PTEN/AKT and suppressor of cytokine signaling (SCOS1)/STAT1. This results in depression of CD8+-T-cell activity and the promotion of CRC growth [115]. The role of CRC-derived sEVs in immune and TME regulation is also well described and summarized in a recent review article by Glass et al. (2022) [116].

#### 2.3.3. Hepatocellular Carcinoma

Immunotherapy is a major breakthrough in the treatment of HCC [117]. Inhibitory regulatory T-cells (Tregs) are central mediators of immune escape in the TME. The exosomal resident circular RNA genetic suppressor element 1 (GSE1) was able to induce Treg expansion via miR-324-5p/TGFBR1/Smad3 signaling [118]. CD8-T cell immune suppression may be further regulated indirectly via specific B-cell subpopulations, which secrete immune inhibitory cytokines. To this end, T-cell immunoglobulin, and mucin domain 1 (TIM1) + regulatory B-cells (Bregs) were shown to release IL-10, exhibiting strong immunosuppressive activity on CD8 T-cells. This phenotype was controlled via HCC-derived sEVs that induce the expression of TIM1 in the Bregs [119].

#### 2.3.4. Pancreatic Cancer

Immunologically cold tumors such as PDAC have a highly immunosuppressive TME which harbors immunosuppressive regulatory T-cells (Tregs), M2-polarized tumor associated macrophages (TAMs), and immature myeloid-derived suppressor cells (iMDSCs) that inhibit functional CD8+ T-cell responses. Additionally, iMDSCs can impede proper antigen presentation by dendritic cells (DCs), or anti-tumor responses by M1-polarized macrophages [9]. In PDAC, sEV-resident PDL-1 levels were reported to be inversely correlated to post-surgical survival [120]. However, direct evidence for immune evasion by the respective sEVs in PDAC is still under investigation.

In conclusion, the reported studies underline that sEV-mediated immune evasion is a critical mechanism in GI cancer. This allows tumor cells to survive and grow in the primary tumor matrix, and also supports tumor embedding and proliferation in pre-metastatic niches.

## 3. sEVs as Biomarkers

Due to their presence in all biological fluids, such as blood, urine, or saliva, sEVs are ideal biomarker platforms for multiplexing analysis—e.g., simultaneous detection of proteins, RNAs, or miRNAs. As a part of personalized medicine approaches, non-invasive diagnostics using liquid biopsies from patient blood samples are currently translated into the clinical routine. Liquid-biopsy-derived sEVs carry a variety of protein, RNA, and DNA cargo as diagnostic and prognostic markers. sEVs have advantages for DNA-mutational profiling when compared to circulating-cell-free-DNA (cfDNA); they provide larger DNA fragment sizes up to 10 kB, which improves the quality of sequencing with tumor mutational panels. [9,121].

As diagnostic markers, liquid-biopsy-derived sEVs allow the differentiation of cancer patients from healthy individuals and other non-cancer diseases. Specific markers have been associated with overall survival (OS), disease-free survival, and tumor stage or disease progression. In addition, chemosensitivity as well as treatment response can be monitored using longitudinal liquid-biopsy-based sEV cargo profiling [122]. In GI oncology, several biomarkers have been described already. In particular, both sEV-derived miRNAs and larger miRNA panels as well as long-noncoding RNAs are used as biomarkers for diagnosis and prognosis [9].

A selection of different sEV-derived markers used in the diagnosis of GI-cancer entities are summarized in Table 1.

## 4. sEVs in Cancer Therapy and Vaccination

### 4.1. Strategies for sEV Engineering for Therapeutic Vehicles

In recent years, research has focused on the therapeutic potential of sEVs and their use in cancer therapy. Since sEVs inherit many physiological characteristics of their originating cells, they are used as cell free therapeutics with comparable potency but better safety profiles. To this end, mesenchymal-stem-cell-derived (MSC) sEVs are currently being investigated for different therapeutic applications due to their anti-inflammatory properties, their regenerative potential, as well as their use as vehicles for tumor therapy [147,148].

sEVs can be loaded with pharmaceutical agents, such as chemotherapeutic drugs or siRNAs for anti-cancer treatment. Therefore, different engineering strategies and technologies have been developed to improve loading efficacy or targeting specificity [147,148,149]. Loading can be differentiated by active and passive mechanisms. Passive cargo loading involves incubation with bioactive agents or drugs, whereby uptake in sEVs is improved by sonication, electroporation, freeze-thaw cycles, or incorporation during extrusion. Active cargo loading can be more specific, and is not as damaging for sEVs through processing. This is usually performed by modifying the sEV-producing cells—e.g., by expression vectors, whereby the ectopically expressed cargo is packaged into sEVs via natural biogenesis pathways. To this end, transgenic fusion proteins with tetraspanins or endosomal resident Lysosome-associated membrane protein 2B (LAMP-2B) are generated, which are actively shifted into sEVs. Another strategy involves fusion proteins, peptides, or nanobodies with a glycosylphosphatidylinositol (GPI) anchor that are incorporated into membranes and sEVs. Alternatively, sEVs may be chemically modified post release. Here, DSPE-or DMPE-PEG lipid ankers are used to present peptides or other surface markers by direct coupling or indirect binding via streptavidin-biotin interaction [147,148,149,150].

Tissue targeting strategies for sEVs involve ectopic expression of binding-peptides as fusion proteins on the sEV surface. To this end, tumor-targeting of sEVs could be improved by LAMP-2B-CRGDKGPDC fusion peptides (iRGD). These sEVs displayed highly effective targeting abilities for αv integrin-positive breast cancer cells in vitro [151,152]. Another targeting strategy is to exploit the natural affinity of specific sEVs for different tissues. Here, mesenchymal-stem-cell (MSC)-sEVs have been used to target different cancer cell populations—e.g., pancreatic cancer cells as described in a study by Zhou and co-workers [153]. They used MSC-sEVs to deploy galectin-9 siRNA and oxaliplatin (OXA) (iEXO-OXA platform). The respective sEVs were shown to significantly improve tumor targeting and drug delivery to the tumor region [151,153].

### 4.2. Engineered sEVs in the Treatment of GI–Cancer

There have been approaches to design therapeutic sEVs in the treatment of most GI cancer entities. In 2017, Kamerkar et al. demonstrated that sEVs derived from mesenchymal cells can be engineered in order to carry si- or shRNAs, which specifically target oncogenic KRAS^G12D^, known to be a key driver mutation in PDAC. Treatment with these iExosomes prolonged OS in mouse models and suppressed PDAC tumor growth [154]. This study is one of the most advanced approaches in the field, even including a clinical phase-1 trial (NCT03608631).

Another study used autologous sEVs from Panc-1 cells loaded with gemcitabine (ExoGem) for treatment of Panc-1 tumors in xenografted mice. Tumor growth was significantly suppressed, resulting in prolonged survival after treatment with ExoGem [155]. Moreover, in PDAC, autologous sEVs were specifically targeted to PDAC cells by modifying their surface with arginin-glycin-aspartatic acid (RGD)-sequences (DSPE-PEG-RGD cloaking), thus delivering paclitaxel as a therapeutic agent [156]. As described above, Zhou et al. utilized bone marrow MSC-derived sEVs to facilitate homing to PDAC cells. The authors further employed an interesting concept to extensively modify the MSC-sEVs with two chemotherapeutic agents, luminal gemcitabine, and surface-bound paclitaxel to improve penetration and therapy performance. Efficacy was demonstrated in orthotopic xenograft models and tumor spheroids [157].

In addition to chemotherapeutic agents or siRNA, other cargos like circular RNAs were applied for treatment of different cancer entities. HEK293-cells were transfected with circDIDO1 circulating RNA to produce modified sEVs, which were targeted to gastric cancer cells using RGD sequences. These sEVs inhibited tumor progression via the miR-1307-3p/SOCS2 axis [158].

Other treatment concepts involve the reversion of chemoresistance. In gastric cancer, high expression of c-Met is associated with poor prognosis and chemoresistance. Accordingly, sEVs were modified with siRNAs to target and deplete c-Met from gastric cancer cells in vitro and in xenografted mice, thereby overcoming tumor-invasive properties and cisplatin resistance [159].

In conclusion, sEV engineering is a promising new research field with the potential to establish innovative treatment options and foster clinical translation.

### 4.3. Use of sEVs as Tumor Vaccines and Immunotherapeutic Agents

Cancer vaccines aim to stimulate a person’s immune system to recognize and attack cancer cells. Unlike traditional vaccines, they are designed to facilitate targeting of tumor cells by the immune system. A cancer vaccine usually contains antigens and adjuvants. Ideally, these antigens are tumor-specific, while adjuvants boost activation of the immune system, both in the absence and the presence of the antigen [160].

Tumor-derived sEVs with tumor-specific antigens were described as cancer vaccines in several studies [161,162,163]. In addition, tumor-derived sEVs contain immunostimulatory molecules, like CD70, CD80, OX40, MHC, or heat shock proteins (HSPs), which directly initiate the innate immune cascade or induce pro-inflammatory cytokine signaling as damage-associated-molecular-patterns (DAMPS) [164,165]. There are initial ongoing clinical trials that use sEVs as cancer vaccines with additional immunostimulatory properties (NCT02657460 and NCT01854866). However, one has to consider the pro-tumorigenic role of the respective tumor-derived sEVs, which can also mediate the establishment of pre-metastatic niches in different organs [21,107].

A second strategy is the use of sEVs from immune cells as anti-cancer vaccines. They contain MHC-I and MHC-II complexes and co-stimulatory molecules, such as CD40, CD80, and CD86. Several studies have used sEVs from dendritic cells to activate anti-tumor immunity [166,167,168]. Other concepts utilize sEV engineering by overexpressing CT40L to induce dendritic cell maturation and boost the immune system. Furthermore, modulation of the CD47– signal regulatory protein alpha (SIRP-p-α) “don’t eat me” signal by sEVs was reported to alter macrophage phagocytosis [169]. sEVs have been additionally modified to express stimulator of interferon genes (STING) agonists, which induces toll-like-receptor (TLR) signaling and interferon response, thereby preventing tumor progression [170].

## 5. Conclusions and Future Perspectives

A better understanding of the underlying mechanisms of sEV biogenesis, secretion, and uptake is crucial for identifying novel targets for clinical translation and therapeutic options. sEVs are major regulators of tumor progression and metastasis in GI cancers. They are also important mediators during the establishment of pre-metastatic niches in different organs, as well as in the regulation of chemoresistance. These features of sEVs have a major translational impact on patients’ survival and prognosis. Therefore, several groups are currently trying to identify clinically applicable inhibitors for sEV biogenesis. However, due to utilization of different biogenesis pathways and cellular adaptation mechanisms, so far, the identification and development of suitable inhibitors is still challenging.

In addition, sEVs have exciting potential as prognostic and diagnostic biomarker platforms in different GI cancer entities.

Moreover, there have been attempts to use sEVs as vehicles for different therapeutic approaches, which are still limited in the GI cancer field. Nevertheless, efforts are made to use sEVs for anti-tumor therapy in the clinical context. To this end, sEV engineering techniques for targeting specific cell populations and cargo loading are extensively investigated. Therapeutic agents loaded in sEVs include chemotherapeutic drugs, like gemcitabine or paclitaxel, as well as siRNAs and specific inhibitors. The most advanced study in this field has been published by Kamerkar et al. for pancreatic cancer. The authors have explored the use of therapeutic sEVs with siRNAs targeting KRAS^G12D^ (iExosomes) to inhibit PDAC tumor growth. Currently, there is even a phase-1 clinical trial (NCT03608631) ongoing.

Thus, sEVs have a great translational and therapeutic potential, which needs to be further developed in GI cancer entities.

## Figures and Tables

**Figure 1 cancers-16-00567-f001:**
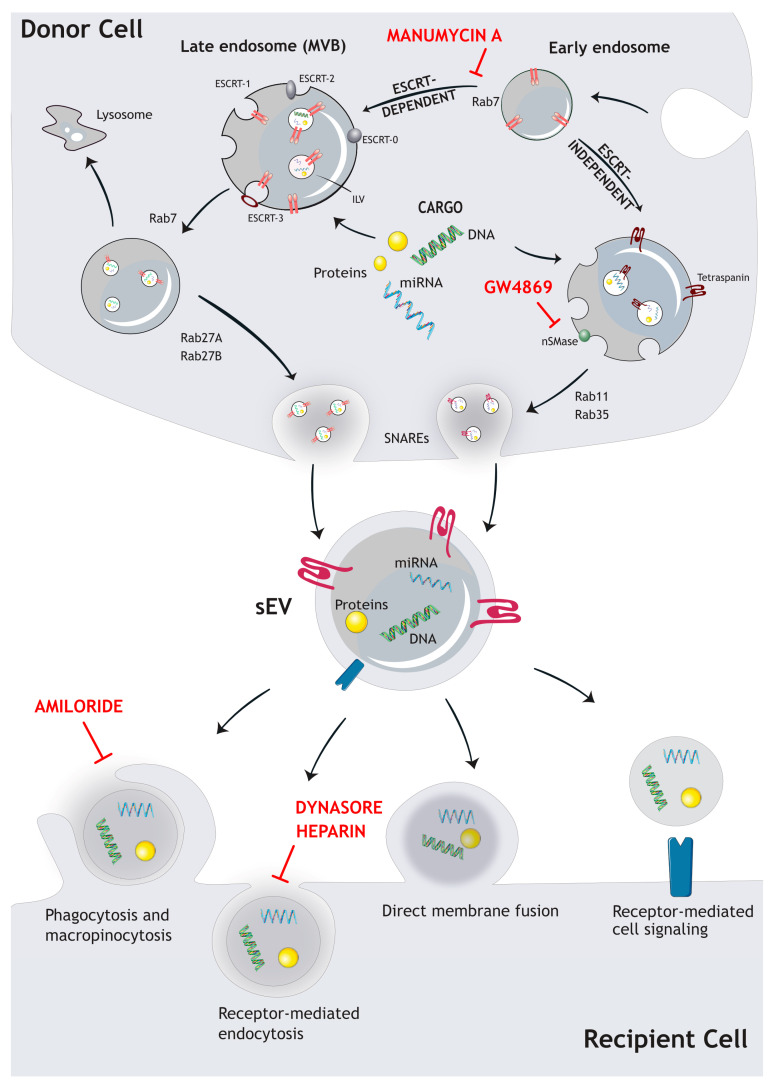
Schematic depiction of pathways involved in sEV biogenesis and uptake in recipient cells. Endosomal sorting complexes required for transport, ESCRT; soluble N-ethylmaleimide-sensitive factor attachment protein receptors, SNARE; Ras associated binding protein, Rab. For description of inhibitors, see Section 1.5.

**Figure 2 cancers-16-00567-f002:**
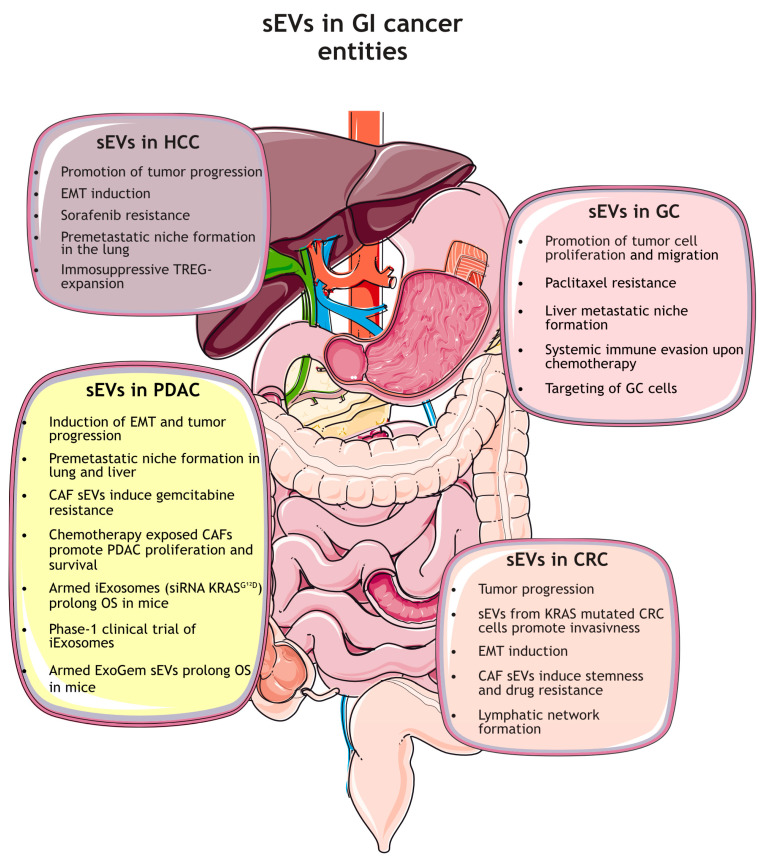
Role of sEVs in gastrointestinal cancer entities. Hepatocellular carcinoma, HCC; Hepatic stellate cells, HSCs; Epithelial-to-mesenchymal-transition, EMT; Inhibitory regulatory T-cells, TREGs; Gastric cancer, GC; pancreatic ductal adenocarcinoma, PDAC; Cancer-associated fibroblasts, CAFs; Overall survival, OS; Colorectal carcinoma, CRC.

**Table 1 cancers-16-00567-t001:** Selection of diagnostic exosomal biomarkers in GI-cancer entities.

	Diagnostic Markers			
Cancer Type	Reference	Exosome Sample/ Clinical Source	Biomarkers	Major Findings
**CRC**	Ogata-Kawata et al. 2014 [123]	Patient serum 88 CRC patients 11 healthy controls	7-miRNAs: let-7a miR-1229 miR-1246 miR-150 miR-21 miR-223 miR-23a	Higher levels of these 7 miRNAs in primary CRC patients Expression was downregulated after surgical resection Verification in vitro
	Ostenfeld et al. 2016 [124]	Patient plasma 2 patient cohorts: Cohort 1: 6 patients with CRC and 5 healthy controls Cohort 2: 7 Stage-III CRC patients (samples were collected prior to surgery and 6 months after)	8 miRNAs: miR-16-5p miR-23a-3p miR-23b-3p miR-27a-3p miR-27b-3p miR-30b-5p miR-30c-5p miR-222-3p	Identification and Isolation of EpCam+ sEVs from CRC patients In the sEVs, miRNA levels were increased and decreased after surgery
	Dong et al. 2016 [125]	Patient serum 76 preoperative CRC patients, 76 matched healthy controls	mRNA KRTAP5-4 mRNA MEGEA3 lncRNA BCAR4	Expression of KRTAP5-4, MAGEA3, and BCAR4 derived from sEVs was increased in CRC patients
	Wang et al. 2017 [126]	Patient plasma 50 early-stage CRC patients 50 matched healthy volunteers	miR-125a-3p	Increased levels in stage I/II CRC patients Diagnostic power of CEA is enhanced by correlating with miR-125a-3p expression
	Barbagallo et al. 2018 [127]	Patient serum 20 CRC patients and 20 healthy controls	lncRNA UCA1, lncRNA TUG1	UCA1 was downregulated, whereas TUG1 was upregulated in CRC patients Combination of UCA1 with TUG1 AUC = 0.814
	Karimi et al. 2019 [128]	Patient serum 25 CRC patients and 13 matched healthy controls	miR-301a miR-23a	high expressions of exosomal miR-23a and miR-301a in CRC patients miR-301a and miR-23a were able to discriminate CRC patients from normal subjects by ROC
	Liang et al. 2019 [129]	Patient plasma 61 CRC patients and 61 healthy controls	lncRNA RPPH1	RPPH1 plasma levels were higher in treatment-naive CRC patients, but lower after tumor resection Exosomal RPPH1 in CRC plasma showed a better diagnostic value (AUC = 0.86) compared to CEA and CA19-9
	Maminezhad et al. 2020 [130]	Patient serum 45 CRC patients and 45 healthy individuals	6 miRNA signatures: let-7a miR-150 miR-143 miR-145 miR-19a miR-20a	Upregulation in CRC patients, but miR-143 and miR-145 were shown to be downregulated
**GC**	Wang et al. 2017 [131]	Patient serum 20 healthy controls and 20 individuals with GC training (90 GC vs. 90 NCs) and blinded phases 20 GC vs. 20 NCs	miR-19b-3p miR-106a-5p miR-17-5p miR-30a-5p	miR-19b and miR-106a were markedly overexpressed in individuals with gastric cancer (GC) compared to healthy controls 2 miRNAs (miR-19b-3p and miR-106a-5p) correctly discriminated 19 out of 20 GC serum samples (95% sensitivity) and 18 out of 20 normal samples (90% specificity) in the blinded phase miRNAs (miR-19b-3p and miR-106a-5p) were related to GC lymphatic metastasis (*p* < 0.01) and expressed at higher levels in stages III and IV compared to I and II stages (*p* < 0.05)
	Li et al. 2018 [132]	Patient plasma 67 gastric cancer patients	miR-217	Expression of 4 miRNA levels of circulating sEVs were altered Exosomal miR-217 was increased in GC patients and the expression was significantly up-regulated in GC tissues In GC, CDH1 has been reported to be a tumor suppressor and to be downregulated CDH1 is a direct target of miR-217 Overexpression of miR-217 enhanced gastric cancer cells proliferation, and reduced exosomal CDH1 level
	Fu et al. 2018 [133]	Patient serum serum samples of 20 gastric cancer patients (14 male and 6 female) and age matched 20 healthy volunteers (13 male and 7 female)	TRIM3 miR-20a	TRIM3 knockdown promoted tumor growth and metastasis of gastric cancer Exosomal TRIM3 was decreased in the serum sEVs of gastric cancer patients sEV-mediated delivery of overexpressed TRIM3 could suppress gastric cancer growth TRIM3 is negatively regulated by miR-20a
	Cai et al. 2019 [134]	Patient Serum 29 healthy people and 63 gastric cancer patients	Lnc RNA PCSK2-2:1	Lnc RNA PCSK2-2:1 expression level in serum sEVs of gastric cancer patients was significantly downregulated expression level of Exo-Lnc RNA PCSK2-2:1 was correlated with tumor size, tumor stage, and venous invasion AUC of Exo-Lnc RNA PCSK2-2:1 was 0.896. At the optimal cut-off value, the diagnostic sensitivity and specificity were 84% and 86.5%, respectively.
	Shao et al. 2020 [135]	96 paired gastric cancer tissues Patient plasma	hsa_circ_0065149	Hsa_circ_0065149 expression was only significantly downregulated in gastric cancer Hsa_circ_0065149 levels were significantly decreased in plasma sEVs of early GC patients
**HCC**	Xue et al. 2019 [136]	80 patients with histologically HCC Patient serum 30 clinical controls	8 miRNAs: miR-122, miR-125b, miR-145, miR-192, miR-194, miR-29a, miR-17-5p, and miR-106a	Significant correlation between serum exosomal miRNAs and tumor size Significant survival difference of HCC patients with high or low exosomal miR-106a
	Gosh et al. 2020 [137]	Normal liver tissue HCC tissue Plasma-derived sEVs	miR-10b-5p/miR-221-3p/miR-223-3p ↑ Distinguishing low alpha-fetoprotein (AFP)	Improving diagnosis in HCC patients with low AFP Identification of liver specific exosomal miRNAs miRNAs were validated in normal and in HCC tissues and in plasma-derived sEVs AFP level was found below 250 ng/mL in about 94% of HCV-HCC and 62% of HBV-HCC patients
	Yao et al. 2020 [138]	Serum of controls and hepatitis, cirrhosis, and HCC patients	Exosome-derived lncRNAs lnc-FAM72D-3 ↑, lnc-EPC1-4 ↓	lnc-FAM72D-3 knockdown decreased cell viability and promoted cell apoptosis lnc-EPC1-4 overexpression inhibited cell proliferation and induced cell apoptosis the expression levels of lnc-EPC1-4 correlated with serum alpha-fetoprotein level lnc-FAM72D-3 and lnc-EPC1-4 might contribute to hepatocarcinogenesis
	Kim et al. 2021 [139]	serum samples from 239 HCC patients and 45 non-HCC patients	miR-125b ↓	Significant downregulation of miR-125b in HCC patients with metastasis Migration and invasion abilities were significantly inhibited by Exosome-mediated miR-125b
	Sun et al. 2021 [140]	HCC patients Patient serum	combination of miR-101 and miR-125b ↓	Combination of miR-101 and miR-125b expression was significantly downregulated in both tissue and serum sEVs of HCC patients Higher diagnostic utility for HCC (area under the curve (AUC) = 0.953) was shown using a combination of miR-101 and miR-125b downregulation
	Wei et al. 2022 [141]	90 HCC patients Patient serum	miR-370-3p ↓, miR -196a-5p ↑	Lower expression of miR-370-3p and higher expression of miR-196a-5p detected in serum sEVs of HCC patients Serum exosomal miR-370-3p and miR-196a-5p were associated with tumor size, tumor grade and TNM stage Overexpressed miR-370-3p or silenced miR-196a-5p suppressed proliferation, invasion, and migration of Huh-7 HCC cells
**PDAC**	Melo et al. 2015 [142]	Patient serum 192 patients 100 controls	Glypican1	GPC1 is a cell surface glycoprotein specifically enriched on cancer-cell derived sEVs With absolute specificity and sensitivity, GPC1+ sEVs could be detected in the sera of pancreatic cancer patients By monitoring and isolating GPC1+ circulating sEVs, healthy controls could be distinguished from patients with early and late stage pancreatic cancer In pre- and post-surgical pancreatic cancer patients, levels of GPC1+ crExos correlate with tumor burden and survival
	Allenson et al. 2017 [143]	Patient plasma 68 PDAC patients of all stages	Mutant *KRAS*	In patients’ plasma, droplet digital PCR (ddPCR) was performed on exoDNA and cfDNA for sensitive detection of KRAS mutants KRAS mutations in exoDNA were identified in 7.4% of localized PDAC, 66.7% of locally advanced, 80% of metastatic PDAC patients, and 85% of age-matched controls, In comparison, mutant KRAS cfDNA was detected in 14.8%, 45.5%, 30.8%, and 57.9% of these individuals, respectively
	Goto et al. 2018 [144]	Patient serum 32 PDAC patients 29 IPMN patients 22 controls	miR–21 miR–451a miR–191	In patients with pancreatic cancer and IPMN-lesions, the expression of exosome-derived miR-191, miR-21 and miR-451a was significantly up-regulated (*p* < 0.05) The diagnostic accuracy of the exosome-derived miRNAs was 5–20% superior compared with the expression of the respective serum bulky miRNAs
	Carmicheal et al. 2019 [145]	Patient serum	Glypican1 EpCAM	In order to identify tumor-specific spectral signatures, label-free analysis of sEVs isolated from normal and pancreatic cancer cell lines was performed using surface enhanced Raman Spectroscopy (SERS) and principal component differential function analysis (PC-DFA) sEVs could be differentiated originating from pancreatic cancer patients or normal pancreatic epithelial cell lines with 90% accuracy
	Lux et al. 2019 [120]	Patient serum 55 patients with PDAC	Exo c-Met Exo PD-L1	By flow cytometry, serum-sEVs from pancreatic cancer patients which were bound to latex beads and stained with antibodies against c-Met or PD-L1 A higher fluorescent signal could be measured for c-Met in PDAC-patients compared to patients with benign disease (*p* < 0.001) By combining the test with levels of the established tumor marker carbohydrate antigen 19-9 (CA 19-9), the diagnostic power could even be improved Pancreatic cancer patients with high levels of PD-L1-positive serum sEVs showed a significantly shorter postoperative survival time (7.8 vs. 17.2 months, *p* = 0.043)
	Reese et al. 2020 [146]	Patient serum	miR-200b; EpCAM+ miR-200c; EpCAM+	From total EpCAM-positive serum sEVs, a biomarker panel consisting of miR-200b and miR-200c showed by receiver operating characteristic (ROC) curve analysis an enhanced diagnostic accuracy of carbohydrate antigen 19-9 (CA.19-9) to 97% (*p* < 0.0001) High expression of miR-200c in total serum sEVs correlated with shorter overall survival (*p* = 0.038) In EpCAM-positive serum sEVs, also high expression of miR-200b correlated with shorter overall survival (*p* = 0.032)

Keratin Associated Protein 5-4, (KRTAP5-4); Melanoma-associated antigen 3, (MAGEA3); Breast cancer anti-estrogen resistance 4 (BCAR4); Carcinoembryonic antigen, (CEA); urothelial carcinoma-associated 1 (UCA1); Taurine Up-Regulated 1, (TUG1); Ribonuclease P RNA Component H1 (RPPH1; Carbohydrate antigen 19-9, (CA 19-9); Cadherin 1, (CDH1); Tripartite motif, (TRIM); Alpha-fetoprotein, (AFP); Hepatitis-C-Virus, (HCV); Hepatitis-B-Virus, (HBV); Tumor node metastasis, (TNM); Glypican1, (GPC1); Intraductal papillary mucinous neoplasm, (IPMN); Epithelial cell adhesion molecule, (EpCAM).

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
