# Peer review of "Emerging Roles of Small Extracellular Vesicles in Gastrointestinal Cancer Research and Therapy"

_cancers, 2024, doi:10.3390/cancers16030567_

Round 1

Reviewer 1 Report

Comments and Suggestions for Authors

Section: 1. Exosomes – biogenesis, cargo loading, secretion & uptake

1.       Line 40, “””” Extracellular vesicles (EVs) are lipid bilayer membrane-nanoparticles that are……….”””. This is not a nanoparticle. Its nanovesicles that behaves as nanoparticle.. It should be Lipid bilayer membrane bound structure, that acts a nanoparticle

2.       Line 42 , such as apoptotic bodies (100- 5000nm),

The size of apoptotic bodies is always greater than 1000nm, not 100-5000nm. Please correct it, you can see the references. Karn et al., (https://www.mdpi.com/2227-9059/9/10/1373).

3.       Is there any difference between sEVs and exossomes, if yes please clarify it. If you think there is no difference, please use only one whether sEVs or exosomes.

4.       Line 66-67, “To this end, the focus for biomarker detection has recently shifted to 66 non-invasive liquid biopsy methods, e.g., analyzing sEVs in patient blood samples”

Its not biomarker detection. It’s a biomarker for the detection of disease, please correct it.  

5.       Section 1.1 Biogenesis mechanisms.  

Line 69-70, sEVs are nanovesicles in a size range from 30-150 nm that are secreted by almost all cell types,,,,,,,,,,,,,, this sentence is repeating, already mentioned above, please remove it.

6.       Figure 1 is not of very quality, please make it clear and attractive.

7.       Section 1.5 Inhibition of sEV release and uptake

Please include the mechanism of different inhibitors for sEVs.   

8.        Section; Conclusion and future perspectives

This is too long, need to be very precise.

Comments on the Quality of English Language

 Extensive editing of English language required

Author Response

Rebuttal-letter

Name: Dr. Nora Schneider/ Dr. Tim Eiseler

Affiliation: University Clinic Ulm, Department for Internal Medicine 1

Email: nora.daiss@uni-ulm.de/ tim.eiseler@uni-ulm.de

Dear Assistant Editor Jovana Bogdanović,

please find enclosed our revised review article: “Emerging Roles of Small Extracellular Vesicles in Gastrointestinal Cancer Research and Therapy” by Schneider et al. to be considered for publication in Cancers. In our manuscript we discuss the current state of research on the role of sEVs as key mediators of intercellular communication, biomarkers and therapeutic effectors in gastrointestinal cancer entities. Due to a dismal prognosis and limited therapeutic options, a better understanding of intercellular communication of the tumor and the local niche, as well as in distant pre-metastatic niche formation is imperative. In order to identify novel treatment strategies, sEVs have been extensively studied in the recent years. Furthermore, sEVs have have gained attention as liquid biopsy biomarker platforms with the potential for multiplexing analysis. In addition, we elucidate the use of sEVs as therapeutic vehicles for treatement or vaccination strategies. Thus, in our manuscript, we offer a comprehensive summary on the role of sEVs in gastrointestinal cancer progression, diagnosis, prognosis and treatment.  

First, we would like to thank all reviewers for their comments and suggestions, which greatly improved our manuscript. In the extensively revised version, we provide all the requested changes, as detailed in the point-by-point discussion below:

Reviewer 1:

  1. Line 40, “””” Extracellular vesicles (EVs) are lipid bilayer membrane-nanoparticles that are……….”””. This is not a nanoparticle. Its nanovesicles that behaves as nanoparticle.. It should be Lipid bilayer membrane bound structure, that acts a nanoparticle.

Author response:

We agree, that our wording was imprecise and have changed the term to nanovesicles (Line 45).

  1. Line 42 , such as apoptotic bodies (100- 5000nm),

The size of apoptotic bodies is always greater than 1000nm, not 100-5000nm. Please correct it, you can see the references. Karn et al., (https://www.mdpi.com/2227-9059/9/10/1373).

Author response:

In line with the reviewer comment, we have changend the size to 1000-5000nm.

  1. Is there any difference between sEVs and exossomes, if yes please clarify it. If you think there is no difference, please use only one whether sEVs or exosomes.

Author response:

We apologize for using inconsistent terms for small extracellular vesicles and have changed all designations to “sEVs”. We also added a definition of sEVs according to MISEV critera (Line 49-55).

  1. Line 66-67, “To this end, the focus for biomarker detection has recently shifted to 66 non-invasive liquid biopsy methods, e.g., analyzing sEVs in patient blood samples” Its not biomarker detection. It’s a biomarker for the detection of disease, please correct it.

Author response:

As suggested, we have rewritten the sentence to improve accesibility for readers (Line 74-78).

  1. Section 1.1 Biogenesis mechanisms. Line 69-70, sEVs are nanovesicles in a size range from 30-150 nm that are secreted by almost all cell types,,,,,,,,,,,,,, this sentence is repeating, already mentioned above, please remove it.

Author response:

As proposed, we removed the sentence.

  1. Figure 1 is not of very quality, please make it clear and attractive.

Author response:

We thank the reviewer for this important suggestion and have revised the design of the figure to make it more attractive. We also improved labeling and added both biogenesis and uptake inhibitors to enhance the impact.

  1. Section 1.5 Inhibition of sEV release and uptake. Please include the mechanism of different inhibitors for sEVs.

Author response:

We thank the reviewer for this comment. As suggested, we complemented this section and added a more detailed description of underlying molecular mechanisms for the different inhibitors (Line 181-201).

  1. Section; Conclusion and future perspectives. This is too long, need to be very precise.

Author response:

As recommended by the reviewer, we have rewritten and shortened this section to make it more concise (Line 574-597).

  1. Comments on the Quality of English Language. Extensive editing of English language required.

Author response:

In line with the recommendation of the reviewer, english language and grammar was improved considerably in our manuscript.

In conclusion, we would like to again thank the reviewers for their valuable comments, which greatly improved accessibility and readability of our manuscript. We believe that this extensively revised version now comprehensively discusses the emerging role of sEVs in gastrointestinal cancer research and therapy.

We therefore believe that our review article will be of interest to the broad readership of Cancers and beyond. The manuscript is not under simultaneous consideration by any other journal. All authors have seen and agreed with the content of the manuscript and declare no competing interests. 

Thank you very much for considering the revised version of our review article.

We are looking forward to hearing from you.

Sincerely yours,

Nora Schneider and Tim Eiseler 

(corresponding author)

Reviewer 2 Report

Comments and Suggestions for Authors

Major comments: This review does not adhere to the MISEV guidelines of classifying vesicles as either small or large EVs. If there is to be a deviation from MISEV, this must be acknowledged within the text.

A discussion of the discovery of exosomes should include all major players. See this review: https://www.ncbi.nlm.nih.gov/pmc/articles/PMC3575527/. 

A discussion of exosome biogenesis should include other proteins other than just the ESCRT machinery.

Major cargo and sEV marker discussion should include Jeppesen et al. Reassessment of Exosome Composition as this definitely characterizes that subset. One example is that MVP is in the non-vesicular fraction, not sEVs. Thery has also published studies comparing large and small EVs that should be cited.

A discussion of sEVs typically is put in the broader context of all secreted mediators, including lipoprotein particles and nanoparticles.

The discussion of CRC sEVs has been elaborated on in a recent review in Gastro (Glass 2022). Please include this reference as it comments on TME modulation and sEVs in CRC.

sEV engineering should include a description of modifying EV surface molecules to hone sEVs to the correct tissues.

Page 8, HCF should be HGF.

Minor comments:

Simple summary: Don’t capitalize “Colorectal cancer” in list after colon. Vital is the incorrect term for “vital mediators”. These errors within the simple summary speak to the larger recommendation of having a content editor look over this review for ensuring clear understanding by readers as well as proper word usage. These changes need to be made throughout the review as there is incorrect use of apostrophes, verb tense, etc.

Figure 2 has some bullet points as sentences and some as attributes. Consistency is needed for this figure.

Discussion of CRC sEVs could include other references regarding mutant KRAS transfer.

Comments on the Quality of English Language

numerous grammatical errors

Author Response

Rebuttal-letter

Name: Dr. Nora Schneider/ Dr. Tim Eiseler

Affiliation: University Clinic Ulm, Department for Internal Medicine 1

Email: nora.daiss@uni-ulm.de/ tim.eiseler@uni-ulm.de

Dear Assistant Editor Jovana Bogdanović,

please find enclosed our revised review article: “Emerging Roles of Small Extracellular Vesicles in Gastrointestinal Cancer Research and Therapy” by Schneider et al. to be considered for publication in Cancers. In our manuscript we discuss the current state of research on the role of sEVs as key mediators of intercellular communication, biomarkers and therapeutic effectors in gastrointestinal cancer entities. Due to a dismal prognosis and limited therapeutic options, a better understanding of intercellular communication of the tumor and the local niche, as well as in distant pre-metastatic niche formation is imperative. In order to identify novel treatment strategies, sEVs have been extensively studied in the recent years. Furthermore, sEVs have have gained attention as liquid biopsy biomarker platforms with the potential for multiplexing analysis. In addition, we elucidate the use of sEVs as therapeutic vehicles for treatement or vaccination strategies. Thus, in our manuscript, we offer a comprehensive summary on the role of sEVs in gastrointestinal cancer progression, diagnosis, prognosis and treatment.  

First, we would like to thank all reviewers for their comments and suggestions, which greatly improved our manuscript. In the extensively revised version, we provide all the requested changes, as detailed in the point-by-point discussion below:

Reviewer 2:

  1. Major comments: This review does not adhere to the MISEV guidelines of classifying vesicles as either small or large EVs. If there is to be a deviation from MISEV, this must be acknowledged within the text.

Author response:

To follow the reviewers advice, we have now included the precise description of small extracellular vesicles (sEVs) according to the MISEV giudelines and clarified that the manuscript is focussed on sEVs and this this term is now used throughout the manuscript (Line 49-57). We apologize if this ascpect was not cummunicated clearly in the initial version.

  1. A discussion of the discovery of exosomes should include all major players. See this review: https://www.ncbi.nlm.nih.gov/pmc/articles/PMC3575527/.

Author response:

We agree with the reviewer that the comprehensive discussion on the discovery of sEVs should include all major players. However, in our review article we focus on the translational aspects of sEVs in gastrointestinal cancers. Therefore, we have opted not to include the full history on the intial discovery and further characterization of sEVs. We rather describe that sEVs have been initially discovered as cellular waste bins (Line 58-67) and underline the current focus on their key function in intercellular communication within the primary tumor matrix or to distant metastatic niches, as biomarkers, or therapeutic vehicles.  

  1. A discussion of exosome biogenesis should include other proteins other than just the ESCRT machinery.

Author response:

As suggested by the reviewer, we have complemented this section with additional descriptions on the ESCRT-independent biogenesis mechanisms via nSmase2 and ceramide, as well as on the SIMPLE pathway (Line 95-106).

  1. Major cargo and sEV marker discussion should include Jeppesen et al. Reassessment of Exosome Composition as this definitely characterizes that subset. One example is that MVP is in the non-vesicular fraction, not sEVs. Thery has also published studies comparing large and small EVs that should be cited.

Author response:

In line with the reviewer suggestion, we have extended the discussion on exosomal cargos and markers and included the reference by Jeppesen et al. resepectively (Line 131-146). Also studies by the Thery group are cited [2-4; 15; 24].

  1. A discussion of sEVs typically is put in the broader context of all secreted mediators, including lipoprotein particles and nanoparticles.

Author response:

As recommended by the reviewer, we have adapted the introduction to more consicely discuss the role and definition of sEVs (see MISEV guidelines) as part of the greater family of extracellular vesicles (Line 45-67).

  1. sEV engineering should include a description of modifying EV surface molecules to hone sEVs to the correct tissues.

Author response:

As proposed, we added a section describing targeting strategies for therapeutic sEVs or the exploitation of natural tissue tropisms for different sEV populations. The paragraph was also amended with an example for the use of mesenchymal-stem-cell-derived sEVs for tumor targeting in pancreatic cancer (Line 505-514).

  1. Page 8, HCF should be HGF.

Author response:

We thank the reviewer for pointing this out and have corrected the typo.

  1. Minor comments: Simple summary: Don’t capitalize “Colorectal cancer” in list after colon. Vital is the incorrect term for “vital mediators”. These errors within the simple summary speak to the larger recommendation of having a content editor look over this review for ensuring clear understanding by readers as well as proper word usage. These changes need to be made throughout the review as there is incorrect use of apostrophes, verb tense, etc.

Author response:

The simple summary has been rewritten and changed according to the reviewers comments. In line with the reviewers suggestion, we have corrected and improved english language and grammar throughout the entire manuscript and hope that the current version is now more accesible for readers. Please find the respective changes in the marked-up version of our manuscript.

  1. Figure 2 has some bullet points as sentences and some as attributes. Consistency is needed for this figure.

Author response:

As suggested, the figure design was improved and bullet point statements were harmonized.

  1. Discussion of CRC sEVs could include other references regarding mutant KRAS transfer.

Author response:

As proposed, an additional reference was added [82].

  1. Comments on the Quality of English Language; numerous grammatical errors

Author response:

As stated above, we have made an effort to considerably improve english language and grammar to enhance readability.

In conclusion, we would like to again thank the reviewers for their valuable comments, which greatly improved accessibility and readability of our manuscript. We believe that this extensively revised version now comprehensively discusses the emerging role of sEVs in gastrointestinal cancer research and therapy.

We therefore believe that our review article will be of interest to the broad readership of Cancers and beyond. The manuscript is not under simultaneous consideration by any other journal. All authors have seen and agreed with the content of the manuscript and declare no competing interests. 

Thank you very much for considering the revised version of our review article.

We are looking forward to hearing from you.

Sincerely yours,

Nora Schneider and Tim Eiseler 

(corresponding author)

Round 2

Reviewer 1 Report

Comments and Suggestions for Authors

Authors address almost all my concerns.  

I have one minor suggestion for my comment No.2.

Line 42, such as apoptotic bodies (100- 5000nm),

The size of apoptotic bodies is always greater than 1000nm, not 100-5000nm. Please correct it, you can see the references. Karn et al., (https://www.mdpi.com/2227-9059/9/10/1373).

Please cite this reference Karn et al., (https://www.mdpi.com/2227-9059/9/10/1373).

Comments on the Quality of English Language

English is fine now. 

Author Response

Dear Reviewer, 

as proposed, we corrected the respective sentence and cited the suggested publication. 

Thank you very much for reviewing our manuscript, 

Sincerely, 

Dr. Nora Schneider and Dr. Tim Eiseler